# Large Language Model (LLM)-Predicted and LLM-Assisted Calculation of the Spinal Instability Neoplastic Score (SINS) Improves Clinician Accuracy and Efficiency

**DOI:** 10.3390/cancers17193198

**Published:** 2025-09-30

**Authors:** Matthew Ding Zhou Chan, Calvin Kai En Tjio, Tammy Li Yi Chan, Yi Liang Tan, Alynna Xu Ying Chua, Sammy Khin Yee Loh, Gabriel Zi Hui Leow, Ming Ying Gan, Xinyi Lim, Amanda Kexin Choo, Yu Liu, Jonathan Wen Po Tan, Ee Chin Teo, Qai Ven Yap, Ting Yonghan, Andrew Makmur, Naresh Kumar, Jiong Hao Tan, James Thomas Patrick Decourcy Hallinan

**Affiliations:** 1Department of Diagnostic Imaging, National University Hospital, Singapore 119074, Singapore; matthew.chan@mohh.com.sg (M.D.Z.C.); calvin.tjio@mohh.com.sg (C.K.E.T.); tammy.chan@mohh.com.sg (T.L.Y.C.); yi_liang_tan@nuhs.edu.sg (Y.L.T.); alynna.chua@mohh.com.sg (A.X.Y.C.); sammy.loh@mohh.com.sg (S.K.Y.L.); gabriel.leow@mohh.com.sg (G.Z.H.L.); mingying.gan@mohh.com.sg (M.Y.G.); xinyi.lim@mohh.com.sg (X.L.); amanda.chookx@mohh.com.sg (A.K.C.); yu.liu@mohh.com.sg (Y.L.); jonathan.tan.wp@mohh.com.sg (J.W.P.T.); ee_chin_teo@nuhs.edu.sg (E.C.T.); yonghan_ting@nuhs.edu.sg (T.Y.); andrew_makmur@nuhs.edu.sg (A.M.); 2Biostatistics Unit, Yong Loo Lin School of Medicine, Singapore 117597, Singapore; qaiven@nus.edu.sg; 3Department of Diagnostic Radiology, Yong Loo Lin School of Medicine, Singapore 117597, Singapore; 4University Spine Centre, University Orthopaedics, Hand and Reconstructive Microsurgery, National University Health System, Singapore 119074, Singapore; dosksn@nus.edu.sg (N.K.); jonathan_jh_tan@nuhs.edu.sg (J.H.T.)

**Keywords:** autonomous artificial intelligence, large language model, spinal instability neoplastic score

## Abstract

**Simple Summary:**

Spinal tumors can cause instability in the spine, and doctors use the Spinal Instability Neoplastic Score (SINS) to decide whether surgery is needed. However, calculating this score can be time-consuming and may vary between doctors. This study evaluates whether a privacy-preserving large language model (LLM) can improve the accuracy and speed of SINS scoring. The model was tested in three ways: on its own, assisting a clinician, and compared to clinicians working without any AI support. The authors aim to show that LLMs can reduce variation, improve efficiency, and support more consistent decision-making for patients with spinal metastases, potentially leading to faster treatment and better care.

**Abstract:**

**Background:** The Spinal Instability Neoplastic Score (SINS) guides treatment for patients with spinal tumors, but issues arise with complexity, interobserver variability, and time demands. Large language models (LLMs) may help overcome these limitations. **Objectives:** This study evaluates the accuracy and efficiency of a privacy-preserving LLM (PP-LLM) for SINS calculation, with and without clinician involvement, to assess its feasibility as a clinical decision-support tool. **Methods**: This retrospective observational study was granted a Domain-Specific Review Board waiver owing to minimal risk. Patients from 2020 to 2022 were included. A PP-LLM was employed to maintain secure handling of patient data. A consensus SINS reference standard was established by musculoskeletal radiologists and an orthopedic surgeon. Eight orthopedic and oncology trainees were divided into two groups to calculate SINS, with and without PP-LLM assistance. LLM-predicted scores were also generated independently of any human input. **Results:** The main outcomes were agreement with the reference standard (measured by intraclass correlation coefficients [ICCs]) and time required for SINS calculation. The LLM-assisted method achieved excellent agreement (ICC = 0.993, 95%CI = 0.991–0.994), closely followed by the LLM-predicted approach (ICC = 0.990, 95%CI = 0.984–0.993). Clinicians working without LLM support showed a significantly lower ICC compared to both LLM methods (0.968, 95%CI = 0.960–0.975) (both *p* < 0.001). The LLM alone produced scores in approximately 5 s, while the median scoring time for LLM-assisted clinicians was 60.0 s (IQR = 46.0–80.0), notably shorter than the 83.0 s (IQR = 58.0–124.0) required without LLM assistance. **Conclusions:** An LLM-based approach, whether used autonomously or in conjunction with clinical expertise, enhances both accuracy and efficiency in SINS calculation. Adopting this technology may streamline oncologic workflows and facilitate more timely interventions for patients with spinal metastases.

## 1. Introduction

The Spinal Instability Neoplastic Score (SINS) is a tool for assessing patients with spinal tumors, guiding decisions about the need for surgical intervention. SINS calculation requires analysis of multiple radiological and clinical components, including tumor location, spinal alignment, bone lesion quality, degree of pain, vertebral body collapse, and posterior element involvement [1]. However, the complexity, interobserver variability, and time constraints, have all presented barriers to SINS implementation in clinical practice [2,3].

Artificial intelligence (AI) such as large language models (LLMs) and other natural language processing (NLP) are being studied as potential clinical support tools [4]. In imaging reports, LLMs have demonstrated promise in automated calculation of the Coronary Artery Disease Reporting and Data System (CAD-RADS) scores but still had some limitations in processing complex data and unstandardized reports [5]. In that study, ChatGPT-3.5, ChatGPT-4o, Google Gemini, and Google Gemini Advanced were used to automatically calculate the CAD-RADS scores from structured radiologist reports as per the reporting guidelines. LLMs have been used to help standardize reporting and facilitate data extraction [6,7,8]. LLMs have also demonstrated higher performance compared to physician groups with and without AI-assistance, showing the potential for autonomous AI in certain contexts [9,10].

AI has the potential to streamline workflows for the calculation of clinical scores like SINS, while also improving accuracy and efficiency. However, careful development of these AI tools are needed to minimize the limitations of AI alone. Clinicians can play a key role in shaping these tools to ensure that they are safe for clinical deployment [11].

Our study will assess the accuracy and efficiency of LLM-predicted and LLM-assisted SINS calculation by trainee doctors, compared against a reference standard of evaluations by musculoskeletal radiologists and an orthopedic spine surgeon. The main outcomes were agreement with the reference standard (measured by intraclass correlation coefficients [ICC] for total score) and time required for SINS calculation. Through evaluating the performance of our institutional privacy preserving-LLM (PP-LLM) in this setting to calculate SINS, we aim to determine the feasibility and its potential as a clinical decision-support tool.

## 2. Materials and Methods

### 2.1. Study Methodology

This retrospective observational study was granted a waiver by the Domain-Specific Review Board (DSRB) due to its low-risk design, as only de-identified data were analyzed. All analyses were conducted using an institutional deployment of a privacy-preserving large language model (Claude 3.5, Anthropic, San Francisco, CA, USA), ensuring secure handling of patient information.

Claude 3.5 was selected as the model because it was available in a privacy-preserving institutional deployment and had demonstrated strong performance in clinical reasoning tasks at the time of study initiation. In this configuration, all inputs and outputs were processed within the hospital’s secure servers, and no data were transmitted externally, thereby ensuring compliance with institutional governance and national Personal Data Protection Act (PDPA) requirements. This is one of the differences when compared to commercial models such as ChatGPT and Google Gemini. This implementation distinguishes our model as ‘privacy-preserving,’ not by the model’s inherent properties, but by the deployment architecture.

Patients with spinal metastases who underwent MRI between January 2020 and December 2022 at the National University Hospital, Singapore, were included in a random selection. Patients with prior spinal surgery or instrumentation, or without complete MRI, CT, or EMR entries (Epic Systems Corporation, Verona, WI, USA) were excluded. All selected patients were 18 years or older. The SINS, outlined in Table 1, consists of six components. In this study, the MRI report was used to document location, alignment, vertebral body collapse and posterior element involvement, while pain was extracted from the EMR entries and bone lesion quality was evaluated using the CT report.

### 2.2. Study Design and Sample

The MRI and CT reports were generated prospectively by experienced, board-certified radiologists specializing in spine oncology, who regularly collaborate in multidisciplinary meetings with spine surgeons, oncologists, and radiotherapists. Only the main body of each radiology report was used for this study, and any concluding statements or partial SIN scoring provided by the reporting radiologists (based solely on MRI or CT criteria without clinical data) were excluded to ensure a consistent, unbiased assessment.

MRI reports followed a standardized format detailing the vertebral level(s) involved, presence of posterior element involvement (unilateral or bilateral), spinal deformity, and degree of pathological vertebral fractures. CT reports emphasized bone lesion characteristics. To evaluate pain history, the radiology request forms and EMR clinical entries (Epic Systems Corporation, Verona, WI, USA) at the time of the MRI and CT requests were retrieved. These entries classified pain as mechanical, occasional, or absent, thereby completing the clinical data required for SINS calculation.

Three experienced readers, a pair of musculoskeletal radiologists (AA and BB, with 3 and 12 years of experience, respectively) and an orthopedic spine surgeon (OSS, with 7 years of experience) used the same MRI, CT, and EMR data that were made available to the LLM in order to establish a reference standard SINS. First, each reader independently evaluated all cases to assess interobserver agreement. Any discrepancies were resolved through discussion, and a consensus SINS was established as the reference standard for subsequent analyses.

A flowchart of the study design is provided in Figure 1. Eight clinicians were recruited to calculate SINS, as they were all involved in the management of patients with spinal metastases and surgical referrals (AC, Orthopedics 2 years of experience; AK IM/Oncology 3 years of experience; GL Orthopedics 2 years of experience; LX Orthopedics 2 years of experience; MY IM/Oncology 3 years of experience; SL Orthopedics 3 years of experience; YL Orthopedics 2 years of experience; JT Orthopedics 1 year of experience). They were divided into two groups of four, with Group A (AC, GL, MY and SL) calculating SINS with AI assistance for a randomly assigned subset of cases, while Group B (AK, LX, YL, and JT) did so for the remaining cases.

Clinicians were recruited from the orthopedics and oncology trainee pool at our institution. All were directly involved in the management of metastatic spine disease and surgical referrals. Recruitment was voluntary, and participants represented a spectrum of postgraduate experience (1–3 years).

The 76 datasets, each representing a unique visit per patient where an MRI spine, CT spine scan, and clinical consultation were performed, were randomly shuffled and then evenly distributed between the two groups. This approach was chosen to ensure a balanced distribution of cases and mitigate potential biases. For the remaining cases, each group calculated the SINS without AI assistance. This approach was intended to minimize any bias from familiarization with cases, as each case contained distinct clinical features. Although a washout period could have been implemented to allow for review of all cases with and without LLM assistance, familiarization with characteristic clinical information was still likely, potentially affecting the accuracy of the assessment, which justified the alternating design.

The PP-LLM was tasked with calculating the SINS using a chain of thought prompt and a few-shot technique. In few-shot learning, the model is provided with a small number of examples (usually one to three) that demonstrate the expected reasoning process. This allows the LLM to generalize the task without requiring extensive training data. The prompt provided to the LLM instructed it to calculate the SINS per spinal level based on the provided clinical history, MRI report, and CT report. The prompt instructed the model through the prompt as follows:Extract pain history from the imaging request forms and/or the most recent EMR clinical entry.Identify lesion location, spinal alignment, vertebral body collapse, and posterior element involvement from the MRI report.Determine bone lesion quality based on the CT report.

The LLM was asked to calculate the highest SINS per case using this stepwise approach, and 10 cases were presented as examples to guide the LLM’s analysis. The model was explicitly instructed not to make up facts during this process and the temperature was set to 0 (reduces stochasticity of the outputs).

Time for SINS assessment was measured in seconds for each case under supervision by the study team, with and without LLM assistance, to evaluate the impact on time efficiency.

### 2.3. Statistical Analysis

The primary outcomes were:Agreement between LLM-assisted SINS and the reference standard.Agreement between clinician SINS (without LLM assistance) and the reference standard.Agreement between LLM-predicted SINS (no human intervention) and the reference standard.The time taken for SINS calculation across these conditions.

Inter-rater agreement for the total SINS Inter-rater agreement for the total SINS was quantified with the single-measures intraclass correlation coefficient (ICC, two-way random, absolute agreement). ICC values were interpreted as poor (<0.50), moderate (0.50–0.75), good (0.75–0.90), and excellent (>0.90) [12].

Agreement for each individual SINS component was assessed with Gwet’s kappa (AC1) to avoid the κ-paradox under skewed category distributions. AC1 values were graded using the widely adopted Landis–Koch scale: slight (<0.20), fair (0.21–0.40), moderate (0.41–0.60), substantial (0.61–0.80), and almost perfect (>0.80) agreement [13].

For time efficiency, differences in scoring time between the conditions were analyzed using the Mann–Whitney U test due to non-normality of the time data. Statistical significance was set at *p* < 0.05. Missing data were excluded pairwise in the analysis, and descriptive statistics (e.g., medians and interquartile ranges [IQRs]) were provided for time measurements.

## 3. Results

### 3.1. Demographics

A total of 60 patients with spinal metastases were included in the analysis (Table 2). A total of 80 MRI spines were performed for the 60 patients, with 4 out of 80 studies excluded due to incomplete MRI, CT or EMR data, leaving 76 studies for analysis. The mean patient age was 63 years (SD ± 9, range 41–83), and just over half were female (32/60, 53.3%). Most patients were of Chinese ethnicity (47/60, 78.3%), and 95% (57/60) had a pre-operative ECOG score of 0–2. Lung tumors were the most common primary (38.3%), followed by breast (20.0%) and prostate (11.7%). Skeletal Oncology Research Group (SORG) classifications showed a predominance of rapid-growth tumors (40.0%). Regarding neurological function, nearly half of the cohort presented with Frankel B or C, highlighting the substantial disease burden.

### 3.2. Reference Standard and SINS Distribution

Three expert readers independently assessed the Spinal Instability Neoplastic Score (SINS) as outlined in the methods (Table 3). Interobserver agreement for the total SINS was excellent, with an intraclass correlation coefficient (ICC) of 0.999 (95% CI: 0.999–1.000). Agreement across individual SINS categories was almost perfect for all, with the lowest agreement observed in the bone lesion quality category (Gwet’s Kappa: 0.975, 95% CI: 0.939–1.000). Perfect agreement (Gwet’s Kappa: 1.000) was achieved for both location and posterior element involvement. Consensus gradings were assigned for all SINS components. Table 4 presents the distribution of reference standard SINS in the study cohort. More than half of the included studies (52.6%) were categorized as indeterminate (SINS 7–12), while 30.3% were classified as unstable (SINS 13–18), and 17.1% were considered stable (SINS 0–6).

### 3.3. LLM-Predicted, LLM-Assisted, and Non-LLM-Assisted SINS Agreement with the Reference Standard

LLM-predicted total scores demonstrated excellent agreement with the reference standard (ICC = 0.990, 95%CI: 0.984–0.993). Clinicians assisted by the LLM achieved an even higher level of agreement (ICC = 0.993, 95%CI: 0.991–0.994). In contrast, clinicians who assessed cases without LLM assistance had a lower, though still excellent, ICC of 0.968 (95%CI: 0.960–0.975), a statistically significant reduction compared to both the LLM-predicted and LLM-assisted approach (both *p* < 0.001) (Table 5 and Figure 2).

LLM-predicted scores showed high agreement across most subcategories. In particular, spinal alignment achieved perfect agreement (Gwet’s Kappa = 1.000), while posterior element involvement also demonstrated almost perfect agreement (Gwet’s Kappa = 0.969, 95%CI: 0.925–1.000). Pain scoring showed a similarly high level of agreement (Gwet’s Kappa = 0.962, 95% CI: 0.910–1.000). However, location (Gwet’s Kappa = 0.917, 95%CI: 0.845–0.990) and bone lesion quality (Gwet’s Kappa = 0.867, 95%CI: 0.772–0.963) exhibited comparatively lower agreement, indicating that these parameters may be more challenging for the LLM to interpret.

LLM-assisted scores were significantly superior compared to non-LLM-assisted scores in the assessment of location (Gwet’s Kappa = 0.958, 95%CI: 0.931–0.986 vs. 0.897, 95%CI: 0.859–0.937, *p* = 0.005), pain (Gwet’s Kappa = 0.991, 95%CI: 0.978–1.000 vs. 0.942, 95%CI: 0.912–0.972, *p* < 0.001), bone lesion quality (Gwet’s Kappa = 0.948, 95%CI: 0.917–0.978 vs. 0.901, 95%CI: 0.862–0.939, *p* = 0.024), spinal alignment (Gwet’s Kappa = 0.996, 95%CI: 0.987–1.000 vs. 0.965, 95%CI: 0.942–0.988, *p* = 0.011), and vertebral body collapse (Gwet’s Kappa = 0.920, 95%CI: 0.884–0.955 vs. 0.856, 95%CI: 0.810–0.903, *p* < 0.001), However, no statistically significant differences (*p* > 0.05) were observed in the assessment of posterior element involvement (Gwet’s Kappa = 0.957, 95%CI: 0.932–0.983 vs. 0.950, 95%CI: 0.922–0.977).

There were no significant differences between subcategories for LLM-Predicted and LLM-Assisted scores except for bone lesion quality (GA = 0.948, 95%CI: 0.917–0.978 vs. 0.867, 95%CI: 0.772–0.963, *p* = 0.04).

On average, LLM-assisted clinicians outperformed those who assessed cases without assistance. These findings highlight that while human scoring remains reliable across most domains, LLM input enhances agreement with the reference standard in key areas, particularly pain assessment and vertebral body collapse.

### 3.4. Time Efficiency with and Without LLM Assistance for SINS Calculation

In terms of efficiency, most readers completed SIN scoring faster with LLM assistance, as shown in Table 6 and Figure 3. For instance, TWPJ experienced the greatest reduction in median scoring time, from 140 s to just 45 s, while G also saw a substantial drop from 89.5 s to 51 s. Across all readers, the LLM-assisted method yielded a median scoring time of 60.0 s (IQR: 46.0–80.0 s), notably lower than the non-LLM-assisted median of 83.0 s (IQR: 58.0–124.0 s), a statistically significant difference (*p* < 0.001). Furthermore, when the LLM generated its own predicted scores without human involvement, it required only about 5 s. Table 7 shows the main aggregated results for SINS agreement and time savings using LLM assistance.

## 4. Discussion

This study assessed the feasibility and effectiveness of an institutional privacy-preserving large language model (LLM) for assisting clinicians with the calculation of the Spine Instability Neoplastic Score (SINS). We found that both the LLM-predicted and LLM-assisted SINS calculations demonstrated excellent agreement with the reference standard (intraclass correlation coefficients [ICCs] = 0.990 and 0.993, respectively). These values were notably higher than the non-LLM-assisted scores (ICC = 0.968). Although the LLM-assisted approach yielded a slightly higher ICC than the LLM-predicted approach, the difference was not statistically significant (*p* > 0.05), highlighting that both methods were highly accurate. Additionally, LLM-assisted scoring showed a significant reduction in calculation time, shortening the median time required by 23 s compared to the non-LLM-assisted approach (*p* < 0.001). Importantly, the fully automated LLM-predicted approach was the fastest by far (Approximately 5 s), providing the greatest potential for time savings in clinical workflows. Although ICC improvements are statistically significant, the absolute differences appear small. However, in aggregate, even modest gains in reliability may enhance confidence in multicenter trials, reduce interobserver variability in research, and support decision-making consistency in high-volume clinical workflows.

Subgroup analyses highlighted how the LLM benefits individual components of the SINS. Vertebral body collapse showed the greatest improvement in agreement compared to the reference standard, moving from non-LLM (GA = 0.856) to LLM-assisted scoring (GA = 0.920), and location and pain assessments also improved substantially (from GA = 0.897 to GA = 0.958, and from GA = 0.942 to GA = 0.991, respectively). Interestingly, although LLM-predicted scores were similar or superior to human-only assessments (e.g., alignment, GA = 1.000; posterior element involvement, GA = 0.969), they did not consistently outperform LLM-assisted scoring for more nuanced categories such as vertebral body collapse or pain. For example, vertebral body collapse requires distinguishing between thresholds of <50% versus >50% height loss, or cases with no collapse but extensive vertebral involvement. These cut-offs are clinically meaningful yet sometimes difficult to determine, especially when radiology reports use terms like “mild” or “significant” compression without quantitative detail. Similarly, pain scoring depends on whether symptoms are mechanical, which requires clear documentation of positional or loading-related exacerbation. In practice, clinical notes often provide limited or ambiguous descriptions, leading to variability in interpretation. Both categories therefore demand more interpretative judgment, which contributes to lower consistency across raters and algorithms compared to other SINS domains. This suggests that while the LLM excels in structured, objective measures, clinician oversight may remain important for more complex or subjective elements.

Targeted enhancements in training data and algorithmic design, potentially through advanced vision-language models that combine text with MRI and CT imaging, may be needed to improve LLM performance in these areas. Further development of vision language models may also help improve efficiency of automated triage, where the SINS could have an automated draft calculation even at the point of completion of the scan, and ready for radiologist review at the time of reporting. These models could even be combined with existing algorithms for epidural spinal-cord compression across both cross-sectional modalities [14,15]

These results align with prior research suggesting that AI-driven tools can enhance decision-making in clinical practice, particularly when evaluating structured tasks such as severity scoring and risk assessment. For example, McDuff et al. [16] similarly reported improved diagnostic accuracy with LLM assistance for diagnostic reasoning. In our analysis, the synergy between human oversight and automated suggestions appeared valuable, as clinicians could validate or refine the LLM recommendations, which mitigated potential AI-driven errors. This human-in-the-loop model is consistent with earlier work indicating that AI–human collaboration often yields superior outcomes compared to standalone AI systems [17,18]. The strong performance of purely LLM-predicted SINS further underscores the capacity for automated triage, in which high-risk or unstable cases could be flagged immediately after radiologic interpretation for expedited review by spine oncology teams. Allowing fully autonomous SINS calculation could lead to the greatest productivity gains, enabling earlier identification and referral of cases to spine oncologists. However, the safety aspect of such an approach relies on adherence to local clinical protocols, which must ensure appropriate oversight and verification by healthcare professionals [11].

However, results have been mixed across other studies. For example, a study by Goh and colleagues [19] found no significant benefit of LLM assistance in diagnostic reasoning, although their study also showed that LLM alone outperformed unaided clinicians. These variations may reflect differences in LLM deployment strategies and prompt optimization, as our study employed carefully designed prompts to maximize output quality. Similar dynamics have been reported in the computer vision domain. Another study by Agarwal and colleagues [9] demonstrated that radiologists underutilized AI input when interpreting chest radiographs, resulting in lower performance with AI assistance compared to AI alone. These findings echo the article by Rajpurkar and Topol [10] that AI may outperform doctors in specific tasks, challenging the assumption that human-AI collaboration naturally yields superior outcomes.

AI has shown promise when deployed independently for high-volume screening tasks, such as identifying normal chest radiographs [20] and assisting mammography screening [21], where AI improved cancer detection while reducing workload. These findings highlight the potential of task-specific AI deployment, where fully autonomous AI handles routine evaluations, and human oversight is reserved for complex cases. However, not all AI triage applications have demonstrated clear benefits. For example, a study by Savage and colleagues [22] reported that an AI triage system for intracranial hemorrhage (ICH) detection did not improve diagnostic performance or reporting turnaround times, emphasizing that the effectiveness of AI integration is highly dependent on the clinical setting and workflow design. These mixed outcomes reinforce the importance of rigorous, task-specific evaluation before clinical implementation.

Our study contributes to this evolving landscape by showing that LLM-assisted and LLM-predicted SINS scoring both outperform conventional methods, with LLM-predicted scoring offering the fastest performance but LLM-assisted scoring providing the highest accuracy, especially for subjective elements. Recent examination-based studies further support the potential of LLMs to meet or exceed trainee-level performance across various medical domains [23]. For example, Watari et al. [24] showed that GPT-4 matched the average score of Japanese medical residents on a nationwide in-training examination, while Zheng et al. demonstrated that a domain-specific ophthalmology LLM achieved performance comparable to senior residents on clinical vignette questions [25].

Despite these promising outcomes, several limitations must be acknowledged in this study. First, this single-center retrospective design may limit generalizability to other clinical settings or patient populations. Structured reporting formats used at our institution may differ from those in other settings. Variability in pain documentation across electronic medical record systems could also affect reproducibility. Our single-center cohort was predominantly composed of Chinese patients with lung cancer primaries, which may limit external applicability. Multicenter validation will be necessary to confirm the robustness of these findings across diverse populations and reporting environments. Second, we did not evaluate how the use of LLM assistance ultimately affects downstream decisions (e.g., surgical planning or referral to radiotherapy), leaving the real-world impact on patient outcomes uncertain. Third, the performance of LLMs in SINS calculation may also be influenced by model size and architecture. More powerful models such as GPT-4o or Gemini 1.5 could potentially achieve superior accuracy. Future studies should examine how scaling influences performance, efficiency, and consistency across different clinical settings. Fourth, the clinicians used in the study may have familiarity with local reporting formats, which may have introduced systemic bias, and results may not fully generalize to clinicians from other institutions or specialties. Last, our dataset excluded cases with incomplete imaging or EMR data, and we did not perform a separate analysis of incomplete or ambiguous reports. While this decision ensured consistency in benchmarking, it limits insight into how LLMs perform under real-world conditions where reports are often partial or incomplete. Future studies should specifically evaluate this subgroup to better reflect practical clinical workflows.

Our dataset excluded cases with incomplete imaging or EMR data, and we did not perform a separate analysis of incomplete or ambiguous reports. While this decision ensured consistency in benchmarking, it limits insight into how LLMs perform under real-world conditions where reports are often partial or incomplete. Future studies should specifically evaluate this subgroup to better reflect practical clinical workflows.

Future work can examine optimal methods for integrating these AI tools into routine clinical practice. For example, automated SINS scoring could be embedded into structured report templates, enabling real-time feedback to clinicians. LLMs could also be used to triage cases by flagging those likely to be unstable, allowing prioritization in busy reporting environments. A human-in-the-loop model, integrated with PACS or EMR systems, may represent a safe and efficient pathway to adoption. Robust governance and accountability frameworks will be essential for clinical deployment. Multicenter prospective studies should validate our findings across broader settings, assess long-term effects on patient outcomes, and explore cost-effectiveness, resource utilization (e.g., initial radiotherapy vs. surgery), and time-to-treatment metrics. Continued investigation of practical implementation and real-world benefits remains essential [26]. Further AI development must involve all stakeholders to overcome the multiple challenges facing AI deployment and research [27,28]. Spine oncologists, in particular, need to steer this transformation to preserve clinical effectiveness and ensure a positive impact on patient care [29,30].

## 5. Conclusions

In conclusion, our findings show that an LLM-based approach, whether fully automated or in a clinician-assisted manner, offers high accuracy and improved efficiency for SINS calculation. By significantly reducing the time required for scoring and enhancing consistency, the technology can potentially streamline oncologic care pathways and expedite critical interventions for patients with spinal metastases. Allowing autonomous SINS calculation by the LLM could further optimize productivity and triage, yet a safe implementation relies on careful integration into local protocols to ensure appropriate oversight. As AI-based tools continue to evolve, their deployment in routine clinical practice could highlight high-risk spinal lesions more promptly and enable more timely, targeted multidisciplinary care.

## Figures and Tables

**Figure 1 cancers-17-03198-f001:**
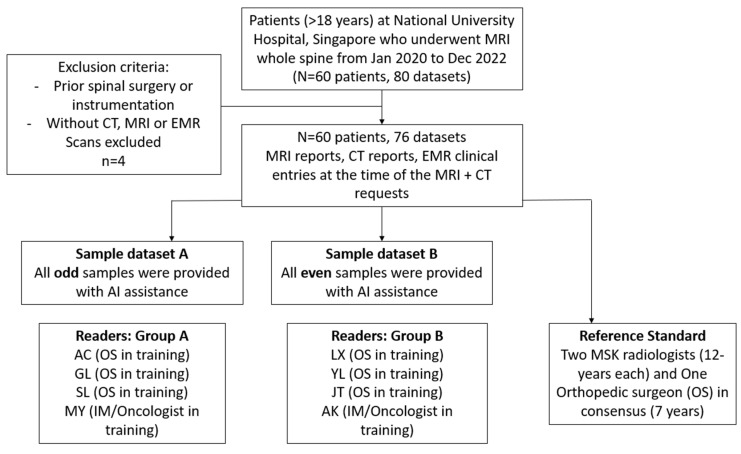
LLM-assisted SINS study flowchart. OS = Orthopedic surgeon. MSK = Musculoskeletal. IM = Internal Medicine. Dataset defined as containing all three criteria during a patient encounter; MRI, CT and EMR clinical entry describing pain.

**Figure 2 cancers-17-03198-f002:**
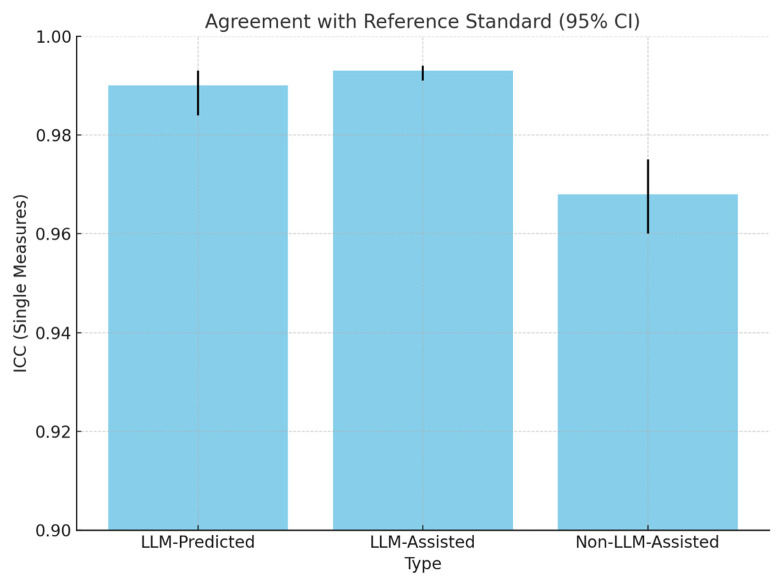
Total SINS agreement with the consensus reference standard.

**Figure 3 cancers-17-03198-f003:**
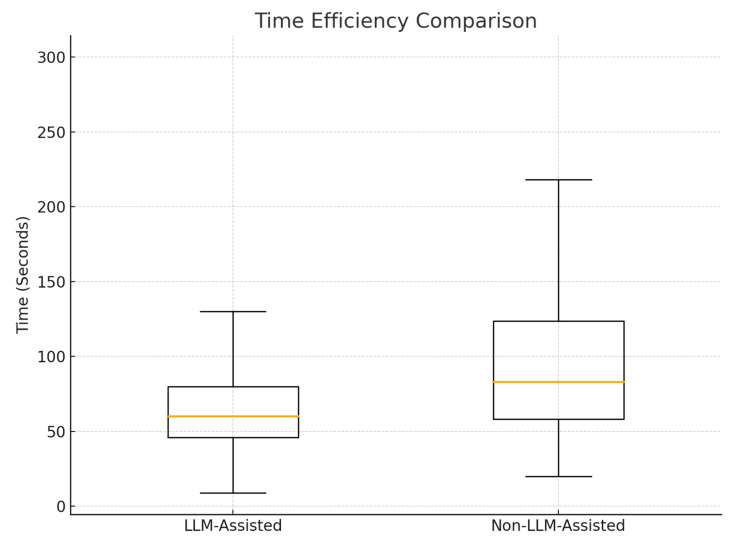
Average time efficiency comparison between LLM-assisted and without LLM-assistance.

**Table 1 cancers-17-03198-t001:** Spinal Instability Neoplastic Score (SINS) categories.

Category	Score	Details
**Location**	3	Junctional (occiput-C2, C7-T2, T11-L1, L5-S1)
	2	Mobile spine (B3-C6, L2-L4)
	1	Semi-rigid (T3-T10)
	0	Rigid (S2-S5)
**Pain**	3	Yes
	1	No (occasional pain but not mechanical)
	0	Pain free lesion
**Bone Lesion Quality**	2	Lytic
	1	Mixed (lytic/blastic)
	0	Blastic
**Spinal Alignment**	4	Subluxation/translation
	2	De novo deformity (kyphosis/scoliosis)
	0	Normal alignment
**Vertebral body collapse**	3	>50% collapse
	2	<50% collapse
	1	No collapse with >50% body involved
	0	None of the above
**Posterolateral involvement of the spinal elements**	3	Bilateral
	1	Unilateral
	0	None of the above
**Interpretation of Total Score** 0–6: No mechanical instability 7–12: indeterminate stability based on the SINS. Patient with who experience mechanical pain generally benefit from stabilization 13–18: mechanically unstable fractures. Stabilization is normally required in consenting patients who are fit for surgery		

Categories adapted from [1].

**Table 2 cancers-17-03198-t002:** Baseline Demographic and Oncological Characteristics of Patients with MESCC.

Characteristics	Dataset (*n* = 60, MRI Spines 76 *)
Mean age (years)	63 (SD ± 9, range 41–83)
Sex	
Male	28 (46.7%)
Female	32 (53.3%)
Race	
Chinese	47 (78.3%)
Malay	10 (16.7%)
Indian	3 (5.0%)
Pre-operative ECOG score	
0–2	57 (95.0%)
3–4	3 (5.0%)
Mean Charlson Comorbidity Index	8 (SD ± 2, range 2–11)
Tumor Subtype (SORG)	
Slow Growth	18 (30.0%)
Moderate Growth	18 (30.0%)
Rapid Growth	24 (40.0%)
Diagnosis	
Known Cancer	46 (76.7%)
New Diagnosis	14 (23.3%)
Cancer Type	
Breast	12 (20.0%)
Lung	23 (38.3%)
Prostate	7 (11.7%)
Renal	2 (3.3%)
Liver	3 (5.0%)
Myeloma/Plasmacytoma Gastrointestinal	3 (5.0%)
Gastrointestinal	6 (10.0%)
Gynecologic	3 (5.0%)
Unknown	1 (1.7%)

Notes—ECOG = Eastern Cooperative Oncology Group; SORG refers to the Skeletal Oncology Research Group classification system used to categorize the tumor subtypes based on their growth rates. * 4 excluded due to incomplete data (incomplete MRI, CT, and/or electronic medical record data).

**Table 3 cancers-17-03198-t003:** Interobserver agreement between three expert readers.

Agreement with Reference Standard	Expert Readers
Total score (ICC)	0.999 (0.999–1.000)
Location (Gwet’s Kappa)	1.000
Pain (Gwet’s Kappa)	0.989 (0.967–1.000)
BLQ (Gwet’s Kappa)	0.975 (0.939–1.000)
SA (Gwet’s Kappa)	0.979 (0.950–1.000)
VBC (Gwet’s Kappa)	0.989 (0.966–1.000)
PEI (Gwet’s Kappa)	1.000

Notes—Location = Location of lesion in the spine, BLQ = Bone lesion quality, SA = Radiographic spinal alignment, VBC = Vertebral body collapse, PEI = Posterior element involvement. The agreement between SINS calculations was evaluated using intraclass correlation coefficient (ICC) for the total score and Gwet’s AC1 (Gwet’s Kappa) for individual category scores.

**Table 4 cancers-17-03198-t004:** Overall Distribution of SINS in the Study Sample.

SINS Range	Number of Cases (n)	Percentage (%)
0–6 (Stable)	13	17.1
7–12 (Indeterminate)	40	52.6
13–18 (Unstable)	23	30.3
Total	76	100%

**Table 5 cancers-17-03198-t005:** Agreement with the reference standard, total score and subsets.

Agreement with Reference Standard	LLM-Predicted Scores	LLM-Assisted Scores	Non-LLM-Assisted Scores
Total Score (ICC)	0.990 (0.984–0.993)	0.993 (0.991–0.994)	0.968 (0.960–0.975)
Location (Gwet’s Kappa)	0.917 (0.845–0.990)	0.958 (0.931–0.986)	0.897 (0.859–0.937)
Pain (Gwet’s Kappa)	0.962 (0.910–1.000)	0.991 (0.978–1.000)	0.942 (0.912–0.972)
BLQ (Gwet’s Kappa)	0.867 (0.772–0.963)	0.948 (0.917–0.978)	0.901 (0.862–0.939)
SA (Gwet’s Kappa)	1.000	0.996 (0.987–1.000)	0.965 (0.942–0.988)
VBC (Gwet’s Kappa)	0.882 (0.795–0.968)	0.920 (0.884–0.955)	0.856 (0.810–0.903)
PEI (Gwet’s Kappa)	0.969 (0.925–1.000)	0.957 (0.932–0.983)	0.950 (0.922–0.977)

Notes—Location = Location of lesion in the spine, BLQ = Bone lesion quality, SA = Radiographic spinal alignment, VBC = Vertebral body collapse, PEI = Posterior element involvement. The agreement between SINS calculations and the reference standard was evaluated using intraclass correlation coefficient (ICC) for the total score and Gwet’s AC1 (Gwet’s Kappa) for individual category scores.

**Table 6 cancers-17-03198-t006:** Time efficiency per reader with and without-LLM assistance for SINS calculation.

Reader	LLM	Median Time	IQR	Lower Limit	Upper Limit
AC	With LLM	59	20.25	50.75	71
	Without LLM	65.5	56.5	60.5	117
AK	With LLM	46	29.25	32	61.25
	Without LLM	46.5	36.75	38.75	75.5
GL	With LLM	51	21.75	43.25	65
	Without LLM	89.5	42.5	64.5	107
LX	With LLM	56.5	16.75	50	66.75
	Without LLM	58.5	29	46	75
MY	With LLM	56.5	40.25	41.25	81.5
	Without LLM	61	37	53	90
SL	With LLM	78	33	63.5	96.5
	Without LLM	100	85	63	148
YL	With LLM	85.5	50.25	64.75	115
	Without LLM	135	54.25	115.75	170
JT	With LLM	45	20	40	60
	Without LLM	140	80	100	180

Notes—Time efficiency is the median with the interquartile range (IQR), AC, Orthopedics 2 years of experience; AK IM/Oncology 3 years of experience; GL Orthopedics 2 years of experience; LX Orthopedics 2 years of experience; MY IM/Oncology 3 years of experience; SL Orthopedics 3 years of experience; YL Orthopedics 2 years of experience; JT Orthopedics 1 year of experience.

**Table 7 cancers-17-03198-t007:** Summary of SINS predictions with and without LLM-assistance.

Outcome	LLM-Predicted Scores	LLM-Assisted Scores	Non-LLM-Assisted Scores
Agreement with Reference Standard for total SINS	ICC = 0.990 (95% CI: 0.984–0.993)	ICC = 0.993 (95% CI: 0.991–0.994)	ICC = 0.968 (95% CI: 0.960–0.975)
Time Efficiency	~5 s	60.0 s (IQR: 46.0–80.0)	83.0 s (IQR: 58.0–124.0)

Notes—The agreement between the reference standard and calculated SINSs was assessed using intraclass correlation coefficient (ICC) for the total score. 95% Confidence Intervals (CI) are provided. Time efficiency is the median with the interquartile range (IQR).

## Data Availability

The data generated or analyzed during the study are available from the corresponding author upon request.

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
