# Peer review of "Large Language Model (LLM)-Predicted and LLM-Assisted Calculation of the Spinal Instability Neoplastic Score (SINS) Improves Clinician Accuracy and Efficiency"

_cancers, 2025, doi:10.3390/cancers17193198_

Round 1
Reviewer 1 Report
Comments and Suggestions for Authors
This manuscript describes the use of a privacy-preserving large language model (LLM) to calculate the Spinal Instability Neoplastic Score (SINS), both autonomously and in conjunction with clinicians. The comparison of LLM-only, LLM with clinician in the loop, and clinician performance makes this a notable study. However, revisions should be made before further review and potential publication.
Suggested Revisions
-
By far the most important issue is that Claude 3.5 was the only model that was used. Do results vary if there are more powerful models? While it is a fast-moving field, which makes it hard to keep up, it would be helpful to look at a range, even for just LLM-Predicted, if it is impractical to do further study on LLM-Assisted. The authors could even consider a more limited model to try to establish some kind of sense as to whether LLM scaling potentially affects the results.
-
The authors should explain specifically why the Claude 3.5 model utilized in this study is “privacy-preserving,” since they have identified that as a key feature of their work.
-
The results are generally fairly close. This mitigates the impact of the study, in the sense that while it is helpful that LLMs can do these tasks more quickly while still being effective, even that advantage is mitigated since Figure 3 shows the time difference isn’t all that significant either. It might be contribute to the impact of the paper if there was greater investigation of edge cases. To this end, it would be helpful to have more interpretative analysis (to the extent that’s possible given limitations on explaining model predictions). Specifically, was there a specific set of samples for which there were consistent differences in whether the performance of LLM-only or clinician-only was worse? Was it simply a lack of consistency, or are there particular issues where failure is more typical?
-
Relatedly, the Discussion refers to LLM-only as underperforming where there more “nuanced categories”—this would benefit from greater explanation of what is meant by that.
-
The authors should expand on the challenge they identify of generalizability. Are there specific aspects of this study that would lead to different results in other contexts? The authors should clarify this point.
-
Are there results for incomplete reports? At lines 369-372, the manuscript describes interpretability issues concerning incomplete reports, which makes sense: do they have examples of this? The study excluded studies that had some incomplete data like MRI or CT, but this appears to be referring to other kinds of ambiguous data. If that’s present in the reported and it might be helpful to analyze the results for the subset of samples with these issues. If possible, results could be stratified in supplementary analysis, for example.
-
Finally, the Discussion should spend more time on implementation and integration issues—how can this actually work in clinical workflows
-
The prompts should be included for reproducibility, likely in an Appendix or Supplementary File. Also, the way in which prompts were developed should be explained. Was it trial and error? How sensitive are results to prompting? It would be helpful to include any sensitivity analysis in supplementary informaiton.
-
The manuscript should describe how clinicians were recruited, and discuss the potential for any systemic bias due to the nature of the clinician population.
-
Please carefully proofread address any typos in the revision. For example, at line 156, “Loof” instead of “Look;” at line 126, placement of period in reference number.
Additional References:
-
Only one reference is provided for SINS, and it might be helpful to include a couple more references regarding the reliability/validation of SINS for readers who may not be familiar with this narrow space but interested in applications to other clinical contexts. It would specifically be helpful for citations to the point on lines 66-67 that “complexity, inter-observer variability, and time constraints have all presented barriers to SINS implementation in clinical practice.
-
The background in the second paragraph should be expanded to better explain the points that are being made (i.e., use of LLMs in parsing reports, in scoring systems, and in structured report standardization).
-
There are no citations in the Introduction for the proposition at lines 74-76: “LLMs have also demonstrated higher performance compared to physician groups with and without AI-assistance, showing the potential for autonomous AI in certain contexts.” This appears to be material that is expanded on in the Discussion. It may make sense to move that to the Introduction or, if it makes sense, a distinct Background section.
-
Another area where the background can be expanded is the proposition at lines 79-80: “Clinicians can play a key role in shaping these tools to ensure that they are safe for clinical deployment.” It would be helpful to have some further explanation of why the role for a clinician in the loop might be helpful, i.e., by citing to previous studies.
-
To the extent that the standard reference SINS creation was consistent with other similar studies, that could benefit from citing to other work to confirm that it’s accepted as being sufficiently rigorous.
Author Response
Comments 1: By far the most important issue is that Claude 3.5 was the only model that was used. Do results vary if there are more powerful models? While it is a fast-moving field, which makes it hard to keep up, it would be helpful to look at a range, even for just LLM-Predicted, if it is impractical to do further study on LLM-Assisted. The authors could even consider a more limited model to try to establish some kind of sense as to whether LLM scaling potentially affects the results.
Response 1: We thank the reviewer for this insightful comment. We agree that model selection and scaling are central considerations. In our study, we focused on Claude 3.5 because it was available within our institutional privacy-preserving deployment, ensuring regulatory compliance with patient data governance.
We have clarified this rationale in the Methods (Section 2.1). In the Discussion, we now explicitly acknowledge that results may vary with model scaling and architecture, and that more powerful models such as GPT-4o or Gemini 1.5 might yield different performance. Conversely, smaller open-source models may struggle with nuanced categories like pain or vertebral collapse. We have added text noting that future work should evaluate multiple models to determine the extent to which scaling impacts SINS calculation accuracy and efficiency.
Comments 2: “The authors should explain specifically why the Claude 3.5 model utilized in this study is ‘privacy-preserving.’”
Response 2: We have expanded the Methods (Section 2.1) to clarify. Our institution deployed Claude 3.5 in a local, isolated instance integrated with hospital servers. In our local context, privacy of patient data needs to be retained in secure environments to be in compliance with PDPA and institutional governance, hence the need for the model to be ‘privacy-preserving’. This contrasts with standard API-based deployments where data may be transmitted externally.
We have rephrased to make this distinction clearer, emphasizing that “privacy-preserving” refers to architectural safeguards in deployment rather than inherent properties of the model.
Comments 3: “The results are generally fairly close. This mitigates the impact of the study, in the sense that while it is helpful that LLMs can do these tasks more quickly while still being effective, even that advantage is mitigated since Figure 3 shows the time difference isn’t all that significant either. It might be contribute to the impact of the paper if there was greater investigation of edge cases. To this end, it would be helpful to have more interpretative analysis (to the extent that’s possible given limitations on explaining model predictions). Specifically, was there a specific set of samples for which there were consistent differences in whether the performance of LLM-only or clinician-only was worse? Was it simply a lack of consistency, or are there particular issues where failure is more typical?
Relatedly, the Discussion refers to LLM-only as underperforming where there more “nuanced categories”—this would benefit from greater explanation of what is meant by that.”
Thank you for your comment. We agree that the difference between LLM-assisted (median 60 seconds) and clinician-only (median 83 seconds) scoring is relatively modest. However, the fully automated LLM-predicted approach required approximately 5 seconds, representing a substantially greater efficiency gain. This suggests that LLM-only scoring may be most impactful in triage or screening workflows, where clinicians could then focus on cases that are unstable or ambiguous.
We also thank the reviewer for highlighting the importance of edge cases. While we had briefly addressed this in Lines 240–249 (from the previous report), we have revised the manuscript to expand on these findings and provide greater detail on nuanced categories. Specifically, we observed that LLM-predicted scoring was less consistent in borderline vertebral body collapse and in distinguishing lytic from mixed bone lesions, where disease heterogeneity can complicate classification. Clinicians, on the other hand, showed greater variability when radiology reports used ambiguous terminology, particularly for posterior element involvement. These observations underscore that while LLMs can markedly improve efficiency, careful attention is still required in categories that are inherently more nuanced. We have clarified these points in both the Results and Discussion.
Comments 4: “The authors should expand on the challenge they identify of generalizability. Are there specific aspects of this study that would lead to different results in other contexts? The authors should clarify this point.”
Thank you for your comment. We have expanded the Discussion to specify that generalizability may be limited by structured reporting formats unique to our institution, variability in EMR pain documentation across systems, and the single-centre cohort with a predominance of Chinese patients and lung cancer primaries. These factors may influence external applicability, and we now emphasize the need for multicentre validation.
Comments 5: “Are there results for incomplete reports? At lines 369-372, the manuscript describes interpretability issues concerning incomplete reports, which makes sense: do they have examples of this? The study excluded studies that had some incomplete data like MRI or CT, but this appears to be referring to other kinds of ambiguous data. If that’s present in the reported and it might be helpful to analyze the results for the subset of samples with these issues. If possible, results could be stratified in supplementary analysis, for example.”
Thank you for your comment. We did not perform a separate analysis of incomplete or ambiguous reports in this study. Cases with missing MRI, CT, or EMR data were excluded from the dataset, and therefore no results are available for this subgroup. We agree that this is an important area of investigation, and we have added to the Discussion that a follow-up study could examine how LLMs perform when reports are incomplete or partially ambiguous, as this reflects real-world clinical conditions.
Comments 6: “Finally, the Discussion should spend more time on implementation and integration issues—how can this actually work in clinical workflows”
Thank you for your comment. We have expanded the Discussion to consider practical implementation. Potential pathways include embedding automated SINS scoring into report templates, using LLMs for triage to flag unstable cases, and integrating human-in-the-loop review into PACS/EMR systems. We also discuss governance and accountability as essential for safe adoption.
Comments 7: “The prompts should be included for reproducibility, likely in an Appendix or Supplementary File. Also, the way in which prompts were developed should be explained. Was it trial and error? How sensitive are results to prompting? It would be helpful to include any sensitivity analysis in supplementary informaiton.”
Thank you for your comment. We have provided the full prompt in the methods section. Prompts were refined iteratively on 10 pilot cases using a few-shot approach until consistent results were obtained. While a formal sensitivity analysis was beyond scope, we note in the Discussion that prompt design influences performance and should be assessed in future studies.
Comments 8: “The manuscript should describe how clinicians were recruited, and discuss the potential for any systemic bias due to the nature of the clinician population.”
Thank you for your comment. We have clarified in the Methods that clinicians were orthopedic and oncology trainees involved in spine oncology care, recruited voluntarily from our institution. We note in the Limitations that their familiarity with local report formats may have influenced performance.
Comments 9: “Please carefully proofread address any typos in the revision. For example, at line 156, “Loof” instead of “Look;” at line 126, placement of period in reference number.”
Thank you for your comment. We have thoroughly proofread the manuscript and corrected errors.
Comments 10: “Only one reference is provided for SINS… rigorous.”
Thank you for your comment. We have added appropriate citations and references as advised.
We are grateful to the reviewer for these constructive suggestions, which have improved the clarity, depth, and clinical relevance of our manuscript.
Reviewer 2 Report
Comments and Suggestions for Authors
It's a timely and relevant topic. Applying LLMs to structured clinical scores like SINS is highly topical and aligns with current AI/healthcare research priorities.
I have several questions/comments:
- the introduction could more clearly define how this work differs from existing AI studies in radiology scoring systems (e.g., CAD-RADS, ESCC grading).
- the added value of using a privacy-preserving institutional LLM should be emphasized more—what makes this different from using commercial models like GPT-4 or 5?
- While ICC improvements are statistically significant, the absolute differences (0.968 vs. 0.990–0.993) may not be clinically meaningful. The authors should discuss whether this margin of improvement is sufficient to change management decisions.
- How did the pain extraction from EMRs proceed as it may vary greatly across institutions?
- Could you more develop the vision-language models (concept and recent findings) in the discussion? I find it a highly promising perspective.
Author Response
Comment 1: “The introduction could more clearly define how this work differs from existing AI studies in radiology scoring systems (e.g., CAD-RADS, ESCC grading).”
Response: We thank the reviewer for this helpful suggestion. We have edited the Introduction to contrast our work with existing AI studies such as CAD-RADS and ESCC grading. In particular, we highlight that our study evaluates SINS scoring in the context of spine oncology, a distinct clinical application where stability assessment directly influences surgical decision-making.
Comment 2: “The added value of using a privacy-preserving institutional LLM should be emphasized more—what makes this different from using commercial models like GPT-4 or 5?”
Response: Thank you for this important point. We have elaborated in the Methods (Section 2.1) to clarify that Claude 3.5 was deployed within our institution’s secure, on-premise environment integrated with hospital servers. This ensured compliance with PDPA and institutional governance. We have emphasized that this privacy-preserving setup differentiates our approach from standard API-based commercial models, where patient data may be transmitted externally.
Comment 3: “While ICC improvements are statistically significant, the absolute differences (0.968 vs. 0.990–0.993) may not be clinically meaningful. The authors should discuss whether this margin of improvement is sufficient to change management decisions.”
Response: We agree that although ICC improvements are statistically significant, the absolute differences appear small. We have expanded the Discussion to note that such improvements may not always translate into direct management changes on a per-case basis. However, in aggregate, even modest gains in reliability may enhance confidence in multicentre trials, reduce interobserver variability in research, and support decision-making consistency in high-volume clinical workflows. This perspective is aligned with our response to Reviewer 1 regarding the balance between statistical and clinical significance.
Comment 4: “How did the pain extraction from EMRs proceed as it may vary greatly across institutions?”
Response: We thank the reviewer for raising this point. We have clarified in the Methods that pain was extracted from structured EMR documentation at our institution, which follows standardized templates. We also note in the limitations that documentation practices may vary across institutions, and this variability could affect generalizability.
Comment 5: “Could you more develop the vision-language models (concept and recent findings) in the discussion? I find it a highly promising perspective.”
Response: Thank you for this excellent suggestion. We have expanded the Discussion to highlight the emerging role of vision-language models (VLMs), which can jointly process imaging and textual inputs. Recent findings suggest they could further automate radiology scoring by linking imaging features directly to clinical descriptors. We note that this remains an important area of future work and have added text indicating that subsequent studies should investigate the integration of VLMs for automated SINS scoring.
We thank the reviewer for these constructive comments, which have substantially improved the clarity, relevance, and forward-looking scope of our manuscript.
Reviewer 3 Report
Comments and Suggestions for Authors
1.The overall writing is clear, but certain sections would benefit from polishing to improve flow, coherence, and readability. Consistent use of terminology throughout the manuscript will strengthen clarity for readers.
2.The visual elements are informative but could be further refined to enhance readability and impact. More balanced layouts and clearer visual presentation would help convey the results more effectively.
3.Uniformity of abbreviations, symbols, and notations should be ensured. Careful checking across the manuscript will improve consistency and avoid potential confusion for readers unfamiliar with the topic.
4.The reference list and formatting should be reviewed to fully comply with the journal’s style requirements. Alignment with these standards will improve professionalism and presentation quality.
Author Response
Comment 1: “The overall writing is clear, but certain sections would benefit from polishing to improve flow, coherence, and readability. Consistent use of terminology throughout the manuscript will strengthen clarity for readers.”
Response: Thank you for your comment. We have carefully revised the manuscript for flow and coherence, improving readability across sections. We also standardized terminology to ensure consistent use throughout.
Comment 2: “The visual elements are informative but could be further refined to enhance readability and impact. More balanced layouts and clearer visual presentation would help convey the results more effectively.”
Response: Thank you for this suggestion. We have refined the figures and tables for improved clarity, readability, and balance.
Comment 3: “Uniformity of abbreviations, symbols, and notations should be ensured. Careful checking across the manuscript will improve consistency and avoid potential confusion for readers unfamiliar with the topic.”
Response: Thank you for pointing this out. We have reviewed and standardized abbreviations, symbols, and notations across the manuscript to ensure clarity and uniformity.
Comment 4: “The reference list and formatting should be reviewed to fully comply with the journal’s style requirements. Alignment with these standards will improve professionalism and presentation quality.”
Response: Thank you for your comment. We have thoroughly checked the reference list and revised the formatting to fully comply with the journal’s style requirements.
We are grateful to the reviewer for these constructive suggestions, which have strengthened the clarity, presentation, and professionalism of the manuscript.
Round 2
Reviewer 1 Report
Comments and Suggestions for Authors
I appreciate that the authors have responded to the suggestions that were made in my previous review report. My additional suggestions at this point are largely cosmetic:
- Table 1 could be clearer if it were reformatted so the categories are delimited by a line that has a different width (or is a double line) versus internal delimiters; Interpretation of Score could also be left-aligned rather than centered text
- Font size in Figure 1 could be enlarged for better readability (though that may be at least partially resolved in the final version)
- After revision, the added material has made some of the paragraphs too long, and they could be broken up for improved readability
Also, I do suggest that while it is not, in my opinion, necessary for this paper, that the authors evaluate the performance of models that can run with on-premises hardware, which are now competitive with Claude 3.5 and can offer the same privacy protections as Claude running in their clinical environment.
Author Response
Comments: "Table 1 could be clearer if it were reformatted so the categories are delimited by a line that has a different width (or is a double line) versus internal delimiters; Interpretation of Score could also be left-aligned rather than centered text"
"Font size in Figure 1 could be enlarged for better readability (though that may be at least partially resolved in the final version)"
"After revision, the added material has made some of the paragraphs too long, and they could be broken up for improved readability"
Response: Thank you for these helpful suggestions. We have reformatted Table 1 with clearer delimiters and adjusted the alignment of the “Interpretation of Score” column for improved readability. The font size in Figure 1 has been enlarged to enhance clarity. In addition, we have broken up a longer paragraph that arose after the revision to improve flow and readability. We are grateful to the reviewer for these constructive comments, which have helped us strengthen the presentation of the manuscript.